# Can We Quantify Aging-Associated Postural Changes Using Photogrammetry? A Systematic Review

**DOI:** 10.3390/s22176640

**Published:** 2022-09-02

**Authors:** Omer Dilian, Ron Kimmel, Roy Tezmah-Shahar, Maayan Agmon

**Affiliations:** 1The Cheryl Spencer School of Nursing, Faculty of Social Welfare and Health Sciences, University of Haifa, Haifa 3498838, Israel; 2Department of Computer Science, Technion Israel Institute of Technology, Haifa 3200003, Israel

**Keywords:** posture, photogrammetry, image processing, aging, aging-associated changes, RGBD sensors

## Abstract

Background: Aging is widely known to be associated with changes in standing posture. Recent advancements in the field of computerized image processing have allowed for improved analyses of several health conditions using photographs. However, photogrammetry’s potential for assessing aging-associated postural changes is yet unclear. Thus, the aim of this review is to evaluate the potential of photogrammetry in quantifying age-related postural changes. Materials and Methods: We searched the databases PubMed Central, Scopus, Embase, and SciELO from the beginning of records to March 2021. Inclusion criteria were: (a) participants were older adults aged ≥60; (b) standing posture was assessed by photogrammetric means. PRISMA guidelines were followed. We used the Newcastle–Ottawa Scale to assess methodological quality. Results: Of 946 articles reviewed, after screening and the removal of duplicates, 11 reports were found eligible for full-text assessment, of which 5 full studies met the inclusion criteria. Significant changes occurring with aging included deepening of thoracic kyphosis, flattening of lumbar lordosis, and increased sagittal inclination. Conclusions: These changes agree with commonly described aging-related postural changes. However, detailed quantification of these changes was not found; the photogrammetrical methods used were often unvalidated and did not adhere to known protocols. These methodological difficulties call for further studies using validated photogrammetrical methods and improved research methodologies.

## 1. Introduction

The global population is growing older. According to United Nations projections, by 2050, the percentage of people aged 65 and over will reach 16%, up from 9% in 2019 [1]. This rapid increase in life expectancy has not been matched by a similar increase in the health of the aging population, which often suffers from various diseases and conditions typical of old age [2,3].

One such problem of increasing concern is changes in posture. Posture can be broadly defined as the orientation of the body in space, or the interrelations between different body parts with respect to the ground [4,5,6]. Inadequate, “stooped” posture is one of the hallmarks of aging and is commonly described in the literature as hyperkyphosis, a shift in the center of mass and an increase in postural sway [7,8,9]. These postural changes are caused by many factors, including a decrease in muscle strength, deterioration of sensory functions, and changes in neural structures, such as the parietal and prefrontal cortices [10,11]. These postural changes are linked to gait and balance difficulties, movement disorders, and falls and are, therefore, a concerning cause for morbidity and mortality among older adults [12]. An early identification of these changes can promote prevention programs and the development of effective interventions.

Posture is usually measured by one of three approaches: direct assessments, radiographic imaging, or photogrammetry. Direct assessments are the most common and traditional methods. They rely on visual observation [13,14], plumbline measurement [15], or the use of a manual tool, such as a goniometer or an inclinometer, that is designed to measure angles in the human body [16]. These methods are economical and can be easily administered in a clinical setting; however, their reliability and interrater validity are particularly low [16,17], making them often inadequate, especially when high-precision measurements are required.

Radiographic imaging is currently the gold standard for evaluating static postural angles. To obtain postural data, X-ray imaging is used to view specific bones and joints, followed by an assessment of anatomical landmarks. Despite its accuracy and reliability, radiographic imaging has some major limitations. The examination is costly, it involves potentially harmful radiation, it requires special equipment and training, and measuring whole-body posture is challenging [18,19].

Photogrammetry, the third approach, is based on computerized image processing that provides values for postural variables [20]. Recent developments in this field have made photogrammetry a more accessible method that allows for the extraction of multiple postural variables using relatively simple instruments, such as digital cameras and analog markers [21,22], stereophotogrammetry (i.e., the use of several cameras to acquire 3D information) [23] or 3D cameras [24]. It can provide a whole-body in vivo postural analysis and may prove useful for linking postural variables and function. In recent years, computerized image processing has allowed for major advancements in various medical fields, both at the microscopic level (e.g., cancer diagnosis and prognosis [25,26]) and the macroscopic level (e.g., surgery [27] and mental health [28]); however, these advancements have yet to become prominent in posture assessment [29,30]. More specifically, it is not yet clear whether it can be used to assess aging-related postural changes; measuring aging-related processes is a pressing problem, as identification of functional markers of aging is crucial to our understanding of the aging process and could serve to predict age-related deterioration and allow for the development of effective interventions [31,32]. Indeed, recent initial studies in this direction showed promising results [33]. To our knowledge, no former review has synthesized existing knowledge regarding photogrammetry use as an aging marker; therefore, the aim of this review is to examine the use of photogrammetry in an attempt to: quantify the postural changes occurring with the aging process; determine what photogrammetrical protocols are used in such assessments; determine what further implementations are needed to facilitate photogrammetrical posture assessment with regard to the aging process.

## 2. Materials and Methods

This systematic review was registered in the PROSPERO international database of registered systematic reviews (CRD42021236064). It follows the Preferred Reporting Items for Systematic Reviews and Meta-Analysis (PRISMA) 2020 guidelines for systematic reviews [34].

### 2.1. Search Strategy and Selection Criteria

We conducted systematic searches of the electronic databases PubMed Central, Scopus, Embase, and SciELO from the beginning of records to March 2021. We used the search terms “posture” AND “photogrammetry”. Details of our search strategy can be found in Appendix A.

Inclusion criteria: (1) participants were aged 60 and older [35], or studies included a comparison between older and young adults; (2) standing posture was measured by photogrammetric means. Excluded criteria: (1) studies not in English; (2) reviews or validation studies.

### 2.2. Data Extraction

Following PRISMA guidelines, two reviewers (R.T.S., O.D.) independently conducted a review of the search results. In case of disagreement between them, they consulted a third, senior researcher and discussed the matter until they reached a consensus.

Data were extracted by one member of the review team who reviewed the characteristics of the studies and their participants, focusing on authors, year of publication, age, gender, and number of participants, and the photogrammetric methods and means (i.e., the system used, angles and variables measured, number of cameras and their positioning, and use of markers, outcome variables, and conclusions).

### 2.3. Data Quality

Studies were quality-assessed using the Newcastle–Ottawa Scale customized for cross-sectional studies [36,37]. In this scale, studies are graded in three rankings: selection, comparability, and outcome, with maximum ratings being 5, 3, and 2, respectively. A complete elaboration of the scale is found in [36].

## 3. Results

Our initial screening yielded a total of 946 citations. After the removal of duplicates, 464 records were screened by two independent reviewers. Eleven papers were designated for full-text reviews by two independent reviewers; of these, three were excluded because they were not in English, and four were excluded because they did not regard older adults. One additional paper was identified in citation searching, and finally, five papers were included in the analysis, as the PRISMA diagram (Figure 1) illustrates.

### 3.1. Study Characteristics

All five reviewed papers [38,39,40,41,42] were published between 2012 and 2019; no papers were found between 2020 and 2022. Four studies (detailed in Table 1) included younger and older participants, while one study [38] included only older women (mean age = 70 ± 8). Two studies were limited to women participants [38,39], and the rest included both men and women. Three studies were from Poland [38,39,42], one was from Germany [41], and one was from the People’s Republic of China [40]. Three of the reviewed studies were published in journals with orthopedic scope [38,41,42] and the other two in journals with geriatric scope [39,40]. Only [40] reported receiving external funding.

### 3.2. Study Design

All five papers used a cross-sectional methodology to assess postural changes occurring with aging. Of these, three compared standing posture of younger and older adult participants [39,40,42]; one study examined the relationship between spinal postural changes and static balance in older women [38]; and another study compared differences in postural responses to leg-length differences between younger and older participants [41]. No data were available in the reviewed studies regarding recruitment methods.

### 3.3. Photogrammetric Methods

Considerable variability was found in two important aspects of posture measurements: the postural variables measured (different angles and lengths) and the system that was used to measure them.

#### 3.3.1. Photogrammetric Systems

Two of the included papers [39,42] used the CQ Elektronik System (CQ Elektronik, Czernica K. Wroclawia, Poland), and one study [41] used the Formetric Rasterstereography System (Diers International GmbH, Schlangenbad, Germany). Both are based on rasterstereography, a method in which a pattern of lines is projected onto the participant’s back, making it possible to measure the distance to various anatomical landmarks [43]. One study [40] used a Nikon J4 camera and lines were drawn at five points (neck, thorax, waist, hip, and knee) on a photograph of participants in a sagittal position, and one study [38] reported the use of a photogrammetric measurement system without specifying which type. None of the included reviewed papers mentioned any references for the validity and reliability of the methods that they used.

#### 3.3.2. Postural Variables Measured

Three of the reviewed papers [38,39,42] measured a similar set of variables, acquired from a posterior–coronal angle using a rasterstereographic system (Appendix A). Two of these measured similar postural angles and inclinations, with a total of 14 variables measured, including kyphosis and lordosis angles and spinal inclinations at different heights [39,42]. The third paper measured 6 of the aforementioned 14 variables: kyphosis and lordosis angles, asymmetry of the lower scapula angle and height, shoulder line inclination angle, and maximum lateral deviation of the spinous processes from the midback line [38].

Of the two remaining studies, one [41] also used a rasterstereographic system while taking a posterior–coronal approach as in the papers described above but measured six, slightly different variables, such as kyphosis and lordosis angles in addition to lateral deviation, pelvic obliquity, angle of pelvic torsion, and surface rotation. It is worth noting that while holding the same names and referring to the same physiological features, the angles in this study were computed with a different method than in the three former ones, as the acute angle, rather than the obtuse one, was taken for physiologic meaning. The last study [40] measured five broadly defined sagittal angles—knee, hip, waist, thorax, and neck—each roughly corresponding to the angle of the body feature in the sagittal plane. A comprehensive description and elaboration of the measured angles and features in the reviewed studies is available in Appendix A.

### 3.4. Age-Related Postural Changes

Four of the reviewed papers [38,39,40,42] showed significant postural differences between older and younger adults (Figure 2). One study [42] found significant differences between the young adult age group (aged 20–25 years, *n* = 70) and the older adult age group (aged 60+, *n* = 70), including an increase in upper body tilt, frontal and lateral deviation from axes, and shoulder tilt. The increases in the depth of thoracic kyphosis and trunk inclination were found to be “the most significant age-related changes in body posture” [42].

One study [39] divided participants into decade-based subgroups and compared a group of young women (aged 20–25, *n* = 130) with a group of older women (aged 60–90, *n* = 130). This division allowed researchers to examine the gradual intensification of postural changes over time and the aging process. A comparison of the two age groups revealed significant differences in angle and depth of thoracic kyphosis, trunk declination, and body tilt but no apparent difference in lumbar lordosis. In women aged 60 years and over, significant differences were found over time in curvature and depth of thoracic kyphosis and scapular asymmetry, as well as in lordotic angle, which decreased with age.

In [40], significant age-related changes in the angles of the neck, thorax, and knees, but not in the waist or hip, were found to occur. The neck angle corresponds to cervical lordosis, and the thorax corresponds to knee–thoracic kyphosis, and the authors interpreted the results as an increase in thoracic kyphosis and cervical lordosis with aging. The waist angle, which was interpreted in this study as a measure of lumbar lordosis, showed no significant change with aging. These changes started in the sixth decade of life and deepened with the aging process.

Lastly, in [38], a flattening of lumbar lordosis and an increase in thoracic kyphosis were associated with a loss of center-of-pressure (COP) balance in elderly women. The fifth reviewed study found no significant differences in the response to simulated leg-length inequalities between younger and older adults [41]. The main significant findings of the reviewed papers are summarized in Figure 2.

## 4. Discussion

The main aim of this review was to determine whether photogrammetry can serve as a valid tool in the quantification of aging-related postural changes. Indeed, the changes described in the reviewed literature were in agreement with current knowledge and consisted mainly of a deepening of thoracic kyphosis [39,40,42], sometimes along a flattening of the lumbar lordosis angle [39]. Furthermore, in [38], these aging-associated changes were found to be correlated with the loss of COP balance in elderly women, indicating further postural implications for static spinal postural changes. Although the postural changes occurring with aging were qualitatively described in the reviewed studies, a detailed quantification of these aging-associated changes was not described, possibly due to some limitations.

Even though most of the reviewed papers illustrated postural changes over time, they are nevertheless cross-sectional. As such, none of them discussed the onset or the potential causes of the documented changes; therefore, we can neither discuss causal associations between postural changes and aging nor understand their development. Furthermore, none of the studies reviewed used an accepted photogrammetrical method that had been previously validated against a gold-standard test. Although such standards exist—for example, the Digital Image-based Postural Assessment (DIPA) protocol [44,45]—studies that employ photogrammetry do not appear to rely on them [30,46]. The use of methods that have not been previously validated is an obstacle to systematizing the results of studies aimed at gaining a better understanding of postural changes that occur with aging. A necessary prerequisite for further use of photogrammetry is to ensure that the methodology is validated against a gold standard.

The diversity in filming protocol and measurement variables among studies is yet another limitation that makes it difficult to summarize and compare results. The studies by Drzał-Grabiec [39] and Wild [41], for example, use rasterstereographic data acquired from a posterior–coronal angle. However, because of differences in their set of variables, these studies were not comparable. Although different filming protocols may serve different purposes, none of the included studies provided a rationale for using one protocol over another. Thus, the heterogeneity of measurements makes it difficult to quantify the main changes occurring with age and compare them.

Some of the methods used in the reviewed papers are not suitable for whole-body postural examination [38,39,41,42] because they rely on a rasterstereography system customized for back angles and lengths. Therefore, they are unfit to study daily positions or the interaction between different postural variables, which is among the main advantages of photogrammetry. There is a trade-off between the accuracy of the photogrammetric system and the distance from the subject, in that filming from a greater distance provides less accurate variables but can capture more parts of the subject’s body. Therefore, a future photogrammetric protocol should take into account the distance and angle of filming.

Lastly, all of the included studies scored between three and six points out of ten on the Newcastle–Ottawa scale adapted for cross-sectional studies [36] because of several key limitations: all of the papers except two [39,42] failed to describe how the research population was sampled; none of the reviewed papers reported how the sample size was calculated; and relevant data on the response rate of subjects invited to participate in the studies are missing. This increases the risk of selection bias, weakens conclusions, and makes it difficult to generalize the data for future research. Gender, as a possible confounding variable, was controlled for in the studies, but other confounding variables, such as functional level, comorbidity, socioeconomic status, and body mass index (BMI), were disregarded by the reviewed studies altogether.

Several limitations may have influenced the conclusions of this systematic review. First, the exclusion of papers not written in English may have eliminated at least three papers whose English title or abstract suggested relevance [47,48,49]. Furthermore, the exclusion of validation studies may have caused bias and defeated the attempt to review all of the available literature, and the heterogeneity and poor quality of evidence of the included studies prevented additional analyses of the available data.

This systematic review adds to the growing body of literature discussing biomarkers of the aging process, a main task of the gerosciences [50]. Photogrammetry, as a measure of whole-body posture, has the potential to serve as an easy-to-implement, economical, and noninvasive functional marker, connected to accelerated aging and functional decline, that can be used for early diagnosis and intervention in aging-related maladies, and initial studies in that direction show promising results [33]. This review shows that photogrammetry’s establishment as such a marker, however, requires an investigation of posture and aging in a broader context—both in the theoretical sense, in our understanding of the links between the two, and in the practical sense, requiring larger studies using refined and accepted methodology. Until such an investigation is performed, photogrammetrical results regarding postural quantification of aging should be interpreted with adequate caution.

## 5. Conclusions

Photogrammetry can be used to assess aging-related postural changes, and changes identified via photogrammetrical means concord with known relations between posture and aging; however, major limitations in past studies prevented precise quantification of these changes. The heterogeneity of measurement methods and their lack of validation is an obstacle for the comparison and assessment of the reviewed literature. Although photogrammetry holds great promise in the field of posture assessment, for a quantification of aging-related postural changes to be possible, strict, validated photogrammetrical protocols are required, alongside future research with larger cohorts, a longitudinal setting, and valid control methodology.

## Figures and Tables

**Figure 1 sensors-22-06640-f001:**
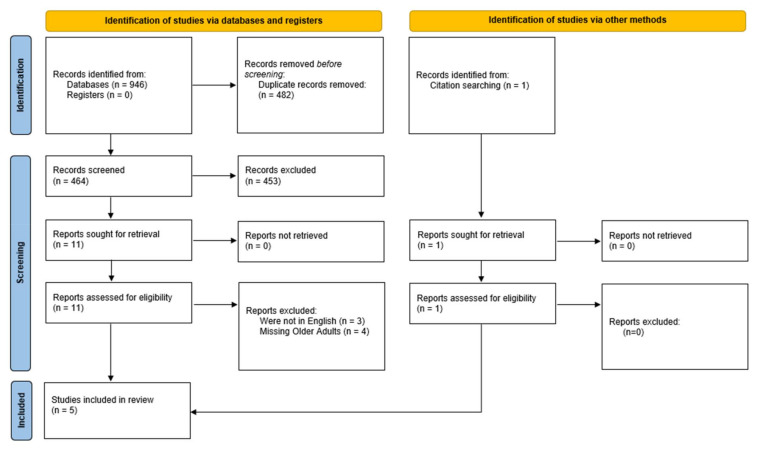
PRISMA diagram.

**Figure 2 sensors-22-06640-f002:**
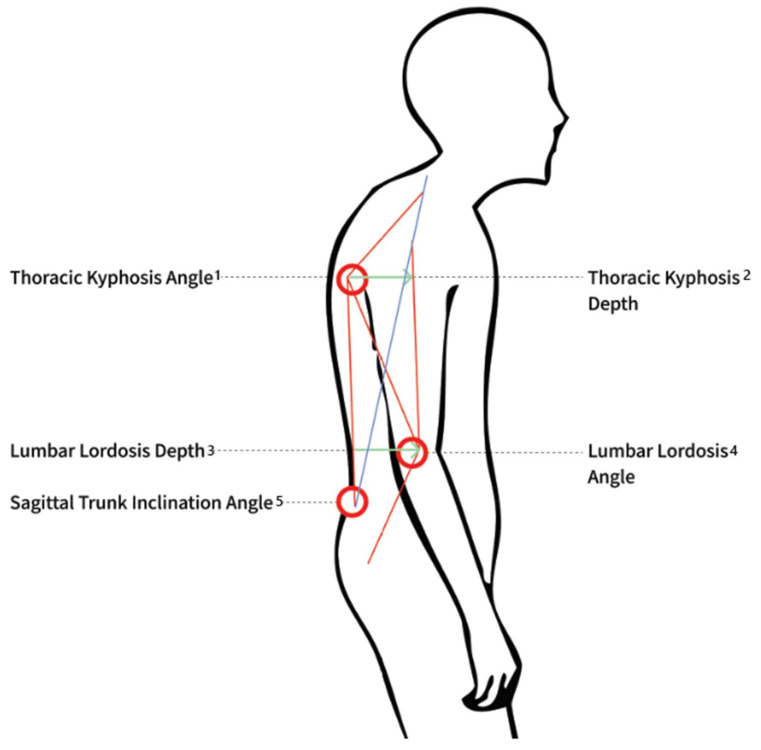
Summary of the main findings: (1) Found in three studies [39,42], named “Thorax” in [40]; (2) Found in three studies [38,42] and [39]; (3) Found in two studies [39,42]; (4) Found in three studies [39,42], named “Waist” in [40]; (5) Found in two studies [39,42]. Bold circles represent angles; green arrows represent depths, measured as sizes, not angles.

**Table 1 sensors-22-06640-t001:** Main characteristics of included studies.

Study	Objective	Sample Characteristics	Measurement Method	System Properties	Ottawa Score
Drzal-Grabiec et al., 2012 [42]	Assess body posture of women and men aged 60 and over.	*n* = 140, 70 participants (35 men and 35 women) aged 60+, 70 participants aged 20–25	CQ Elektronik System	A rasterstereographic system that projects straight lines on the patient’s back to yield measurement deviation. The system is composed of one camera and a projector. Data processing is then carried via a specialized software.	S = 3 C = 1 O = 2
Drzal-Grabiec et al., 2013 [39]	Evaluate parameters that characterize the posture of women aged 60 and over compared with a control group; determine the dynamics of body posture changes in the following decades.	*n* = 260, 130 women aged 60–90, control group of 130 women aged 20–25	CQ Elektronik System		S = 3 C = 1 O = 2
Wild et al., 2014 [41]	Investigate age-related differences in spinal and pelvic responses to simulated leg-length inequalities.	*n* = 107, participants divided into three age groups: 20–39 (mean = 23.8 ± 3), 40–59 (mean = 48.9 ± 5.7), >60 (mean = 68 ± 4.7)	Formetric Rasterstereography System		S = 2 C = 1 O = 2
Drzał-Grabiec et al., 2014 [38]	Assess the influence of age-related changes in spinal curvature on postural balance in elderly women.	*n* = 90, women, mean age = 70 ± 8.01 years	Not mentioned		S = 1 C = 0 O = 2
Gong et al., 2019 [40]	Assess the parameters of standing body posture in the global sagittal plane; determine the dynamics of changes in standing body posture that occur with age and the differences between men and women.	*n* = 226, participants aged 20–89, divided into 7 decade-based age groups	Nikon J4 Camera	Standard photographs were taken; lines were drawn on them to signify postural axes, and angles were measured using computerized software.	S = 2 C = 1 O = 2

## Data Availability

Not Applicable.

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
