# Peer review of "Can We Quantify Aging-Associated Postural Changes Using Photogrammetry? A Systematic Review"

_sensors, 2022, doi:10.3390/s22176640_

Round 1

Reviewer 1 Report

This article is interesting but it's conclusions are not what I was expecting, unfortunately. There are 2 figures and 1 table that helped to understand what is in the text.

The use of photogrammetry to attempt to quantify the postural changes that occurs while people age is very interesting, and the idea of trying to use computer vision to analyze this data is very promissing and will be the future, from my point of view.

This literature review presents a result opposite to what is expected when one begins to read the article, and I believe that the authors when started this investigation weren't expecting this outcome either.

Despite of this, it is an interesting article and a possible base for further investigations in this area, focusing on the comparison of this type of approach (photogrammetry) with the top techniques currently used nowadays.

Overall, after some minor corrections it is an interesting review.

Errors/questions:

Line 119: Table 1 for some reason appears cut off in the document, this must be corrected.

Line 139: "None of the reviewed papers have mentioned any references for the validity and reliability of the methods they used" - Unfortunately this would have been a very important factor that would give much more value to your article, are you sure you didn't find anything in the 482 articles that did this?

Line 153: "the angles in this study were computed in a different method than in the three former ones" - It is necessary to explain the difference in the methods.

Line 184: "The main significant findings are described in figure 2." - They are not described, they are identified and referenced. Furthermore the description should be in the text.

Line 185: "This section may be divided by subheadings. It should provide a concise and precise description of the experimental results, their interpretation, as well as the experimental conclusions that can be drawn." - This was obviously left here by mistake and has to be removed.

Line 197: "Even so, detailed quantification of these aging-associated changes was not described, possible due to some limitations of the reviewed studies." - This, in my opinion, is one of the negative aspects of your article, although the studies analyzed themselves are the cause of this. This sentence here demeans the work done by you, it should be revised to try to present the positive points of this analysis

Line 200: "Although most of the reviewed papers illustrated postural changes over time, they are nevertheless cross-sectional. As such, none of them discussed the onset and potential causes of the documented changes, and therefore we can neither discuss causal associations between postural changes and aging nor understand the nature of these postural changes."  - This statement suggests that you were not able to find any relevant information in the analyzed articles or extract any useful data?? This must be revised and corrected.

Reviewer 2 Report

With regard to the manuscript: Can We Quantify Aging Associated Postural Changes Using Photogrammetry? A Systematic Review, submitted to Sensors.

The paper will contribute to knowledge and is worthy of publication, however, the authors should be more detailed when writing the methodological cautions in conducting the photogrammetry (This technique is or not performed by a well-trained executor, time of day since cold or heat could affect postures….). The general scope of the study appears to be acceptable and is of interest, but allow me to give you a few suggestions.

Introduction

·  The introduction provides an overview on aging, posture, three approaches (direct assessments, radiographic imaging, and photogrammetry), but the novelty of the study could be more highlighted.

·  The authors do not explain the reason why they decided to indagate on photogrammetry. Why emphasis is placed on photogrammetry ? Why not was studied the visual observation (using goniometers which are easily obtainable)? To demonstrate that photogrammetry is better than visual observation a comparison between these approaches must be exhibited in introduction. Please add some reference to support this.

Methods

·  Describe the profile of studies in more detail.  

·  From screened 464 records, why only eleven papers were designated ? The lost is very huge.

Results

·  Table 1 is very large, making understanding difficult. Table 1 could be more concise (more focused on numbers, averages). Another table with detailed information could be published as supplementary file, which is intended to be an addition to the main manuscript.

·  Legend of figure 2 must be more explained.

·  The authors described the following issues: Study Characteristics, Study Design, Photogrammetric Systems, Postural Variables Measured and Age Related Postural Changes. I strongly recommend the insertion of more results about the following topics: 1) Included articles have been published in peer-reviewed different journals, 2) The total number of countries of the first author’s affiliation for the included articles, 3) Among the articles included, how many have received funding. Among the articles included, how many did not report the number of healthy older participants, 4) Among the articles included, how many reported the educational level, nationality, or ethnicity of participants.

·  About data collection method, it is important describe WHEN? (Time of day, season of data collection, Data collection duration), WITH WHOM? or HOW? (Of the articles included, the most frequently used recruitment technique was through visits to local organizations and/or groups of older adults, Interviewers/researchers were trained or had training in conducting tests, It was reported whether stops or breaks were taken during the tests?).

Discussion

·  Didactics would improve with the inclusion a figure in discussion section (some scheme drawn by the authors) explaining their findings.

·  It is well written showing interesting interpretation. It is important to reader to know why photogrammetry could be useful for science, for the medical context. Thus, the authors should warn readers to be cautious when interpreting and conducting photogrammetry. I would like this point of view to be more in-depth.

Reviewer 3 Report

Thank you for giving me the opportunity to review the manuscript. The systematic review paper is about postural changes using photogrammetry in older adults. The topic is important; however, there are some issues regarding the manuscript as follows:

1)    Abstract: It should state “PRISMA” in this section.   

2)    Methods:

-       The PRISMA guidelines were used to review all studies in the review. However, the authors failed to mention, which version was used and its need to be cited.

-       Please provide more details on how to use the Newcastle-Ottawa Quality Assessment Scale to evaluate the quality of methodologies based on three parameters and how to indicate scores.

-       Why was the search from gray literature did not include in this review? Please explain.

3)    Results:

-       “All five reviewed papers [38–42] were published between 2012-2019, no papers were 114 found between 2019-2022.” I think Gong et al. [40] published their paper in 2019. Please recheck.

-  Table 1 should be reorganized based on the journal guidelines.  Also, all abbreviations should be noted under the table.

4)    Discussion:

-       “DrzaÅ‚-Grabiec et al., 2014 [38]” Please follow the journal guidelines.

-       Full sentence should be mentioned for the first time, then, followed by abbreviations (e.g, COP), which is very important for the reader to follow.

-       Please mention the strengths and practical implications of this study.

5)    In the supplementary material, the authors should provide the search strategy in the different databases based on postural changes using photogrammetry in older adults.

Round 2

Reviewer 3 Report

I would like to accept this present form. Congratulations!